# Impact of Hamstring Tightness on Muscle Activation in Healthy Young Adults

**DOI:** 10.3390/jfmk10040363

**Published:** 2025-09-23

**Authors:** Eduardo Guzmán-Muñoz, Camila Zurita-Leiva, Felipe Gómez-Araya, Yeny Concha-Cisternas, Antonio Castillo-Paredes, Felipe Montalva-Valenzuela, Rodrigo Yáñez-Sepúlveda, Emilio Jofré-Saldía, Iván Molina-Márquez, Manuel Vasquez-Muñoz

**Affiliations:** 1Escuela de Kinesiología, Facultad de Salud, Universidad Santo Tomás, Talca 3460000, Chile; camilazurita@santotomas.cl (C.Z.-L.); felipegomez@santotomas.cl (F.G.-A.); 2Pedagogía en Educación Física, Facultad de Educación, Universidad Autónoma de Chile, Talca 3460000, Chile; 3Vicerrectoría de Investigación e Innovación, Universidad Arturo Prat, Iquique 1100000, Chile; 4Grupo AFySE, Investigación en Actividad Física y Salud Escolar, Escuela de Pedagogía en Educación Física, Facultad de Educación, Universidad de Las Américas, Santiago 8370040, Chile; antonio.castillo@udla.cl; 5Escuela de Entrenador en Actividad Física y Deporte, Facultad de Ciencias Humanas, Universidad Bernardo O’Higgins, Santiago 8370040, Chile; felipemontalva95@gmail.com; 6Faculty Education and Social Sciences, Universidad Andrés Bello, Viña del Mar 2520000, Chile; rodrigo.yanez.s@unab.cl; 7School of Medicine, Universidad Espíritu Santo, Samborondón 092301, Ecuador; 8Escuela de Ciencias de la Actividad Física, Facultad de Ciencias de la Rehabilitación y Calidad de Vida, Universidad San Sebastián, Santiago 8370040, Chile; emilio.jofre@uss.cl; 9Escuela de Educación Física, Facultad de Educación, Universidad Adventista de Chile, Chillán 3780000, Chile; ivanmolina@unach.cl; 10Programa Doctorado en Ciencias de la Actividad Física, Universidad Católica del Maule, Talca 3460000, Chile; 11Centro de Observación y Análisis de Datos en Salud, Facultad de Medicina y Ciencias de la Salud, Universidad Mayor, Santiago 8580745, Chile; manuel.vasquez@umayor.cl; 12Escuela de Medicina, Facultad de Medicina y Ciencias de la Salud, Universidad Mayor, Santiago 8580745, Chile

**Keywords:** hamstring tightness, electromyography, muscle activation, flexibility, Nordic curl, neuromuscular function, young adults

## Abstract

Background: Hamstring tightness is highly prevalent in young adults and may negatively affect neuromuscular performance. Despite growing interest in the biomechanical and neuromuscular consequences of reduced flexibility, few studies have examined its effect on muscle activation in healthy individuals. This study aimed to compare thigh muscle activation during functional tasks in healthy young males with and without hamstring tightness. Methods: Thirty healthy male participants (18–26 years) were assigned to two groups based on the Active Knee Extension test: normal flexibility (<20°) and hamstring tightness (≥20°). Surface electromyography (sEMG) was used to assess the activation (%MVIC) of the biceps femoris and semitendinosus muscles during four functional exercises: unilateral standing knee flexion, unilateral bridge, elastic-band knee flexion, and Nordic curl. Independent *t*-tests were used to compare muscle activation between groups. Results: Participants with hamstring tightness showed significantly lower activation of the semitendinosus during the unilateral bridge (*p* = 0.036) and Nordic curl (*p* = 0.024). Additionally, biceps femoris activation during the Nordic curl was reduced in the tightness group compared to the normal group (*p* = 0.044). No significant differences were observed in other exercises. Conclusions: Hamstring tightness is associated with reduced activation of key posterior thigh muscles during exercises that require high eccentric or isometric demands. These neuromuscular alterations may impair performance and increase the risk of injury in functional tasks. Clinically, assessing and addressing hamstring flexibility may support muscle recruitment efficiency and injury prevention strategies in young physically active populations.

## 1. Introduction

The hamstrings constitute a large and functionally important muscle group located in the posterior compartment of the thigh [1]. This group includes the biceps femoris (long and short heads), semitendinosus, and semimembranosus muscles. Except for the short head of the biceps femoris, all originate from the ischial tuberosity of the pelvis and insert distally on the tibia and fibula, crossing both the hip and knee joints [1]. This biarticular configuration allows the hamstrings to support knee flexion, hip extension, and pelvic and lower limb stabilization during activities such as walking, running, and maintaining posture [1,2].

To fulfil these biomechanical roles effectively, the hamstrings must exhibit adequate strength, flexibility, and neuromuscular coordination to support coordinated and efficient movement across the hip and knee joints [1,2]. Among these attributes, flexibility plays a pivotal role. Flexibility is the capacity of the muscle–tendon unit to elongate passively, enabling joint movement across the full physiological range without discomfort or structural restriction [3]. Notably, this capacity does not depend exclusively on the muscle fibers themselves but also on the integrated contribution of passive structures such as tendons, ligaments, joint capsules, fascia, and surrounding connective tissue [4]. Consequently, a reduction in hamstring flexibility may stem from changes in any of these components, reflecting a multifactorial limitation that can compromise movement efficiency, increase mechanical stress on adjacent joints, and alter neuromuscular control [5].

Epidemiological evidence indicates that a reduction in hamstring flexibility or “tightness” is highly prevalent in the general population. In early adulthood, hamstring tightness has been reported in approximately 68% of students from a university in India [6]. Occupational data further support these findings: 83.4% of industrial workers involved in prolonged seated tasks aged 18 to 60 years exhibited marked hamstring shortening [7]. Similarly, 55.5% of administrative workers demonstrated hamstring tightness, a condition increasingly associated with extended sedentary periods and the aging process [8].

Among the various clinical tools available for assessing hamstring flexibility, the Active Knee Extension (AKE) test stands out as one of the most reliable and widely used methods in both research and clinical practice [9,10]. This test involves positioning the individual in a supine position with the hip flexed at 90°, while the contralateral limb remains extended and stabilized [11]. From this position, the participant actively extends the knee, and the angle between the thigh and the lower leg is measured [11]. The degree of knee flexion at the point of maximum extension reflects the extensibility of the hamstring muscles. A value of less than 20° is generally considered normal, whereas higher angles indicate limited flexibility or hamstring tightness [12,13].

Given the high prevalence of hamstring tightness and its potential impact on lower-limb biomechanics, it becomes essential to explore how this condition may influence neuromuscular function. Surface electromyography (sEMG) is a non-invasive and widely accepted method for evaluating the electrical activity of muscles during voluntary movements, offering valuable insight into parameters such as activation amplitude, onset timing, and intermuscular coordination [14,15]. This technique has proven valuable for analyzing motor unit behavior in both healthy and altered muscular conditions [16,17,18]. Studies have shown that hamstring tightness can lead to significant changes in activation patterns, including delayed onset and increased recruitment of synergistic muscles [19,20]. These neuromuscular alterations may compromise movement efficiency, increase mechanical load, and raise injury risk [16].

Despite the increasing evidence linking hamstring tightness to altered neuromuscular patterns, few studies have specifically examined the impact of reduced flexibility on muscle activation in young healthy populations [17,19,20]. Most existing studies have focused on injured athletes or indirect indicators of neuromuscular performance, while evidence using sEMG to investigate activation patterns in healthy young adults with hamstring tightness remains scarce. This gap limits our understanding of how flexibility restrictions may be associated with measurable electrophysiological differences in muscle activation. Therefore, this study aimed to compare thigh muscle activation between healthy young adults with and without hamstring tightness. We hypothesized that young adults with hamstring tightness would present differences in electromyographic activation of the semitendinosus and biceps femoris muscles during functional tasks compared with individuals with normal hamstring flexibility.

## 2. Materials and Methods

### 2.1. Study Design

This was an observational, descriptive, cross-sectional study [21]. The dependent variable was the percentage of muscle activation of the biceps femoris and semitendinosus muscles, while the independent variable was the presence or absence of hamstring tightness.

### 2.2. Participants

The study included 30 healthy male young adults between 18 and 26 years of age, enrolled at Universidad Santo Tomás, Talca (Chile). Participants were selected through a non-probability convenience sampling approach and allocated into two groups based on their hamstring flexibility status, as determined by the Active Knee Extension (AKE) test. Group 1 consisted of individuals with reduced hamstring flexibility, while Group 2 included those with normal flexibility.

The minimum required sample size was estimated using the two-tailed *t*-test for independent means. Based on previously published data on sEMG in individuals with and without muscle tightness, a large effect size (Cohen’s d ≈ 1.38) was calculated [17]. Using an alpha level of 0.05 and a statistical power of 90%, the estimated sample size was 12 participants per group. Therefore, a total sample of 30 subjects (15 per group) was considered sufficient to ensure robust statistical power while accounting for potential variability in sEMG measurements.

All participants met the inclusion criteria, which required being male and within the defined age range. Participants were excluded if they presented any inflammatory, painful, or musculoskeletal condition at the time of evaluation, if they had experienced any lower-limb musculoskeletal injury within the previous six months, if they were actively engaged in competitive sports, or if they had performed moderate-to-vigorous physical activity within the 72 h before testing. According to self-report, all participants engaged in recreational physical activity approximately 2–3 times per week, but none had participated in structured resistance training or competitive sports during the previous six months.

Prior to data collection, all participants were informed of the objectives, procedures, risks, and benefits of the study. Each individual voluntarily signed an informed consent form in accordance with the ethical principles outlined in the Declaration of Helsinki.

### 2.3. Procedures

All experimental procedures were conducted in the Somatosensory and Motor Research Laboratory at Universidad Santo Tomás (Talca, Chile). The laboratory environment was thermally controlled and maintained at a temperature of 22 °C. All assessments were performed between 15:00 and 17:00 h.

Hamstring flexibility was assessed using the AKE test. Each participant was positioned in a supine posture on an examination table, with the non-test limb fully extended and relaxed [11]. The test limb was placed in 90° of flexion at both the hip and the knee, stabilized with a wooden support [11]. To ensure proper positioning and prevent compensatory movements, the evaluator manually stabilized the distal femur throughout the test. Participants were then instructed to actively extend the knee as far as possible while maintaining the 90° of hip flexion [11]. The degree of knee extension was measured using a standard electrogoniometer (Pasco^®^, Santiago, Chile) [22]. The fulcrum was aligned with the lateral epicondyle of the femur, the stationary arm with the femoral shaft, and the moving arm with the fibular shaft [22]. Only the dominant limb was assessed based on participant self-report, and the resulting knee angle was used for classification purposes. A knee flexion angle greater than or equal to 20° was considered indicative of reduced hamstring flexibility [12,13]. Based on this criterion, participants were allocated into two groups: a tightness group (≥20° knee flexion) and a normal flexibility group (<20° knee flexion).

Neuromuscular activity was measured using a wireless surface electromyography system (Delsys Trigno™ Lite, Boston, MA, USA). Prior to electrode placement, the skin over the dominant lower limb was shaved, disinfected with alcohol, and marked according to anatomical landmarks following the SENIAM guidelines [23]. The muscles analyzed were the biceps femoris and semitendinosus. For the biceps femoris, the electrode was placed at the midpoint between the ischial tuberosity and the lateral epicondyle of the tibia, while for the semitendinosus, the electrode was positioned at the midpoint between the ischial tuberosity and the medial epicondyle of the tibia [23]. Electrodes were fixed using double-sided adhesive tape to minimize motion artifacts, and participants were placed in the prone position on an examination table during electrode placement.

To normalize EMG signals, a maximal voluntary isometric contraction (MVIC) was first performed. Participants were instructed to contract their hamstring muscles with maximal effort for five seconds while lying prone, with the knee flexed to approximately 30°, supported by a quadriceps bench. Each participant performed three MVIC trials for each muscle, with two minutes of rest between attempts.

After the MVIC, each participant completed a series of four functional exercises in the following order: unilateral standing knee flexion (held for 3 s), unilateral bridge (supine position with the dominant leg in support, held for 3 s), elastic band knee flexion (prone position with resisted movement held for 3 s), and Nordic hamstring curl (kneeling, with a controlled trunk descent lasting approximately 3 s, guided by verbal cues from the evaluators to ensure proper pacing) (Figure 1). Each exercise was preceded by a demonstration and a familiarization trial to ensure correct execution. Three valid repetitions of each exercise were performed, with one minute of rest between repetitions, and sEMG data were collected during all repetitions.

Electromyographic signals were acquired and processed using EMGworks^®^ software version 4.8.0 (Delsys Inc., Boston, MA, USA). Raw signals were sampled at 2148 Hz, band-pass filtered between 20 and 450 Hz, full-wave rectified, and smoothed using a fourth-order zero-lag Butterworth low-pass filter with a 10 Hz cut-off frequency [16,18,24]. For MVIC trials, the EMG signal was analyzed using the root mean square (RMS) method over a stable 2 s window extracted from the central portion of the 5 s contraction phase [25], in order to minimize transient onset and offset effects. A mean RMS value was calculated for each trial, and the highest of these three mean RMS values was then used as the reference for normalization of the EMG signals recorded during the functional tasks.

For each exercise repetition, the signal was analyzed using the mean RMS method over a stable 2 s window extracted from the central portion of the 3 s contraction phase [25]. The first and last 0.5 s were excluded to eliminate transitional artifacts. A mean RMS value was calculated for each repetition, and the average of the three repetitions was used for analysis. The final RMS values for each exercise and muscle were then normalized to the MVIC reference and expressed as a percentage. Normalized EMG values were calculated as the percentage of the maximal reference obtained from MVIC, according to the following formula:




N
o
r
m
a
l
i
z
e
d

E
M
G
%
=
M
e
a
n

R
M
S

o
f

f
u
n
c
t
i
o
n
a
l

t
a
s
k
H
i
g
h
e
s
t

m
e
a
n

R
M
S

f
r
o
m

M
V
I
C
×
100



This procedure allowed standardized comparison of muscle activation across participants and conditions.

### 2.4. Statistical Analysis

All statistical analyses were performed using GraphPad Prism version 8.0 (GraphPad Software, San Diego, CA, USA). Descriptive statistics were computed for all variables and are presented as means ± standard deviations. Data normality was assessed using the Shapiro–Wilk test. To compare the percentage of muscle activation (%MVIC) between the tightness group and the normal flexibility group, independent-samples *t*-tests were conducted for each muscle and exercise, assuming normal distribution and homogeneity of variances. Statistical significance was set at *p* < 0.05. Effect sizes were calculated using Cohen’s d for *t*-tests and interpreted as small (0.2), medium (0.5), or large (≥0.8).

## 3. Results

A total of 30 participants, evenly divided into two groups, were evaluated in this study. The characteristics of the sample are presented in Table 1.

Biceps femoris activation did not differ between groups in three of the four functional tasks (Figure 2). During unilateral standing knee flexion, the normal-flexibility group showed a mean activation of 8.08 ± 1.77%MVIC, whereas the hamstring-tightness group reached 10.63 ± 5.22%MVIC (*p* = 0.058). In the unilateral bridge, values were 15.87 ± 9.11%MVIC and 15.58 ± 10.82%MVIC for the normal-flexibility and hamstring-tightness groups, respectively (*p* = 0.943). Similarly, during elastic-band knee flexion, activation was 22.69 ± 7.21%MVIC in the normal-flexibility group versus 25.66 ± 13.06%MVIC in the hamstring-tightness group (*p* = 0.242). In contrast, the Nordic curl elicited significantly higher biceps femoris activation in participants with normal flexibility (57.97 ± 14.29%MVIC) compared with those presenting hamstring tightness (47.90 ± 14.07%MVIC; *p* = 0.044).

Activation of the semitendinosus differed between flexibility groups in two of the four functional tasks (Figure 3). During unilateral standing knee flexion, the normal-flexibility group exhibited a mean activation of 20.11 ± 8.50%MVIC, whereas the hamstring-tightness group reached 21.34 ± 9.51%MVIC (*p* = 0.367). In the unilateral bridge, semitendinosus activation was significantly greater in participants with normal flexibility (21.96 ± 8.78%MVIC) than in those with hamstring tightness (16.12 ± 6.41%MVIC; *p* = 0.036). No between-group difference emerged during elastic-band knee flexion, where values were 43.73 ± 16.48%MVIC and 39.99 ± 16.72%MVIC for the normal-flexibility and hamstring-tightness groups, respectively (*p* = 0.289). Finally, the Nordic curl elicited the highest overall semitendinosus activity, with the normal-flexibility group displaying markedly greater activation (84.18 ± 23.92%MVIC) than the hamstring-tightness group (66.24 ± 18.31%MVIC; *p* = 0.024).

## 4. Discussion

The present study investigated the influence of hamstring tightness on the activation of thigh muscles during functional exercises in healthy young males. The main findings indicate that individuals with reduced hamstring flexibility exhibited lower activation of the semitendinosus muscle during the unilateral bridge and Nordic curl exercises. Additionally, during the Nordic curl, the biceps femoris activation was significantly greater in the normal-flexibility group.

These results align with previous evidence showing altered activation patterns in individuals with restricted flexibility. For instance, Emami et al. [19] reported that athletes with prior hamstring strain displayed changes in lumbo-pelvic muscle recruitment during hip extension. Similarly, Iguchi et al. [20] demonstrated that compensatory activation of synergistic muscles during hip-dominant tasks could increase injury risk in athletes with reduced flexibility. Our results extend this knowledge by highlighting specific reductions in semitendinosus and biceps femoris recruitment in healthy subjects without previous injury history but with clinically significant hamstring tightness. While prior studies have mostly examined injured athletes or indirect indicators of neuromuscular performance, our findings emphasize that even in non-injured individuals, hamstring tightness may be associated with measurable alterations in muscle activation. By using surface EMG under controlled conditions, we objectively suggest that limited flexibility may represent not only a mechanical restriction but could also be associated with electrophysiological alterations in neuromuscular behavior. This highlights both the preventive and rehabilitative value of assessing and addressing hamstring flexibility as a modifiable factor influencing neuromuscular efficiency.

One plausible explanation for the reduced activation in the tightness group may involve biomechanical disadvantages. Limited hamstring extensibility could alter the length–tension relationship of muscle fibers, particularly during end-range hip or knee movements. This suboptimal mechanical positioning could reduce motor unit recruitment efficiency during eccentric or isometric contractions, especially in exercises such as the Nordic curl, which demand high muscle control [5,13]. Furthermore, fascial stiffness and increased passive resistance may contribute to early fatigue or motor inhibition, diminishing effective activation [4].

Another possible mechanism may involve altered proprioceptive input associated with muscle shortening. It has been proposed that reduced muscle length may diminish the neural drive to the affected muscles, leading to motor control alterations such as reciprocal inhibition, delayed onset of activation, or compensatory overactivation of synergistic muscles [26,27]. Although the precise mechanisms remain uncertain, muscle shortening could influence proprioceptive feedback mediated by muscle spindles, resulting in suboptimal neuromuscular responses to functional demands. Evidence supports this idea: Smilde et al. (2016) demonstrated that changes in the length of neighboring muscles—mediated by myofascial connectivity—can modulate the firing of muscle spindles, indicating that proprioceptive input is sensitive not only to the length of the target muscle but also to passive mechanical interactions [28]. Furthermore, Doguet et al. (2017) found that during eccentric contractions, corticospinal excitability and intracortical inhibition are modulated by muscle length, suggesting a complex integration of proprioceptive signals and descending motor control [29]. While these phenomena have been primarily described in other contexts—such as delayed activation of the tensor fasciae latae and gluteus medius in individuals with iliotibial band tightness [17]—they may also provide a partial explanation for the reduced semitendinosus and biceps femoris activation observed in participants with hamstring tightness in our study. These considerations should be regarded as potential mechanisms rather than definitive conclusions, and they underscore the need for further research before drawing direct inferences about neuromuscular control or injury risk.

From a clinical and functional standpoint, these neuromuscular alterations may have significant implications. Suboptimal activation of key hamstring muscles during demanding tasks could compromise dynamic knee stability, increase the reliance on non-optimal synergists (e.g., gluteus maximus or gastrocnemius), and potentially elevate the risk of muscle strain or joint overload during sport-specific or daily activities [17,19,20]. Therefore, identifying and addressing hamstring tightness may be crucial in preventive and performance-focused programs, particularly among young adult populations frequently exposed to prolonged sitting or participation in sports.

It is important to acknowledge several limitations in our investigation. First, only male participants were included, which limits generalizability to female populations. Second, the assessment was cross-sectional, which does not allow for establishing cause–effect relationships; therefore, the findings should be interpreted strictly as associations. Third, we assessed only two hamstring muscles (semitendinosus and biceps femoris), which provides a focused but limited view of neuromuscular control. Although these muscles are clinically relevant and commonly reported in the literature, functional tasks such as the unilateral bridge and Nordic curl also involve synergists, including the gluteus maximus, gastrocnemius, and erector spinae. Future studies should therefore expand the muscle set to capture a more comprehensive picture of neuromuscular coordination. Fourth, although participants refrained from physical activity during the 72 h prior to testing, other factors that may influence EMG signals—such as residual fatigue, sleep quality, hydration, or nutritional status—were not controlled. Lastly, future longitudinal or interventional studies are needed to determine whether treating hamstring tightness improves neuromuscular activation and reduces injury risk during functional tasks.

## 5. Conclusions

In this study, healthy young males with hamstring tightness showed lower semitendinosus activation during the unilateral bridge and Nordic curl, as well as reduced biceps femoris activation during the Nordic curl, compared with peers with normal flexibility. These findings suggest an association between limited hamstring flexibility and differences in neuromuscular activation patterns during tasks with high eccentric or isometric demand.

## Figures and Tables

**Figure 1 jfmk-10-00363-f001:**
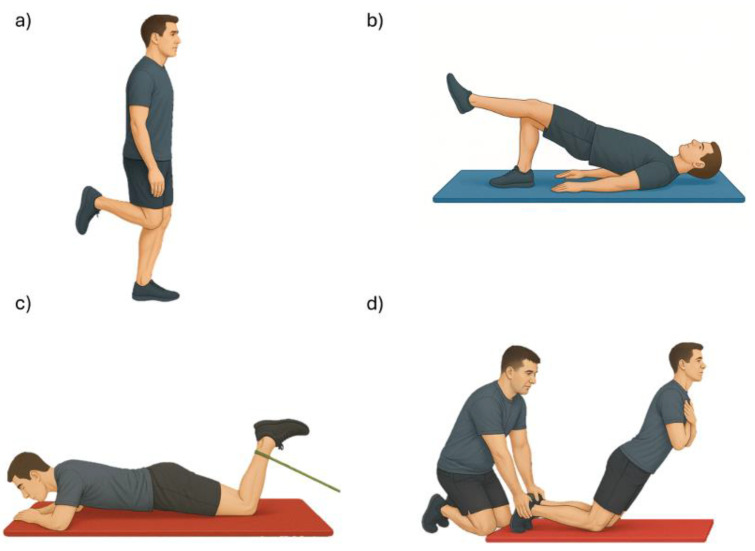
Exercises assessed with surface electromyography: (**a**) unilateral knee flexion, (**b**) unilateral bridge, (**c**) elastic-band knee flexion, and (**d**) Nordic curl.

**Figure 2 jfmk-10-00363-f002:**
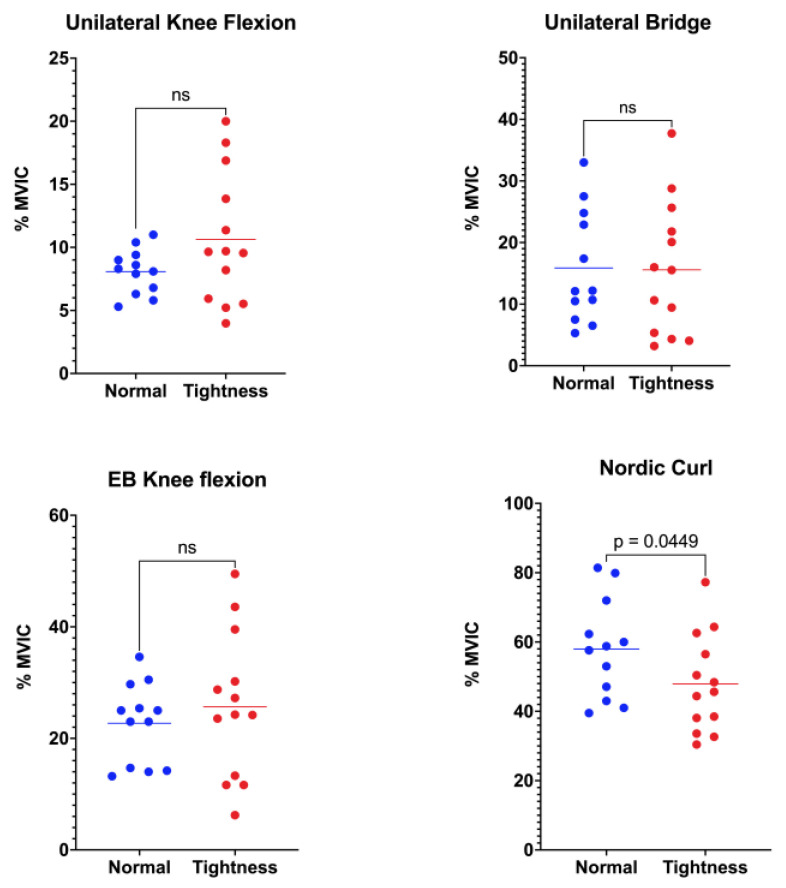
Comparison of biceps femoris muscle activation between the normal-flexibility and hamstring-tightness groups during the following exercises: unilateral knee flexion, unilateral bridge, elastic-band (EB) knee flexion, and Nordic curl. Statistical differences are indicated in the figure; “ns” denotes non-significant differences.

**Figure 3 jfmk-10-00363-f003:**
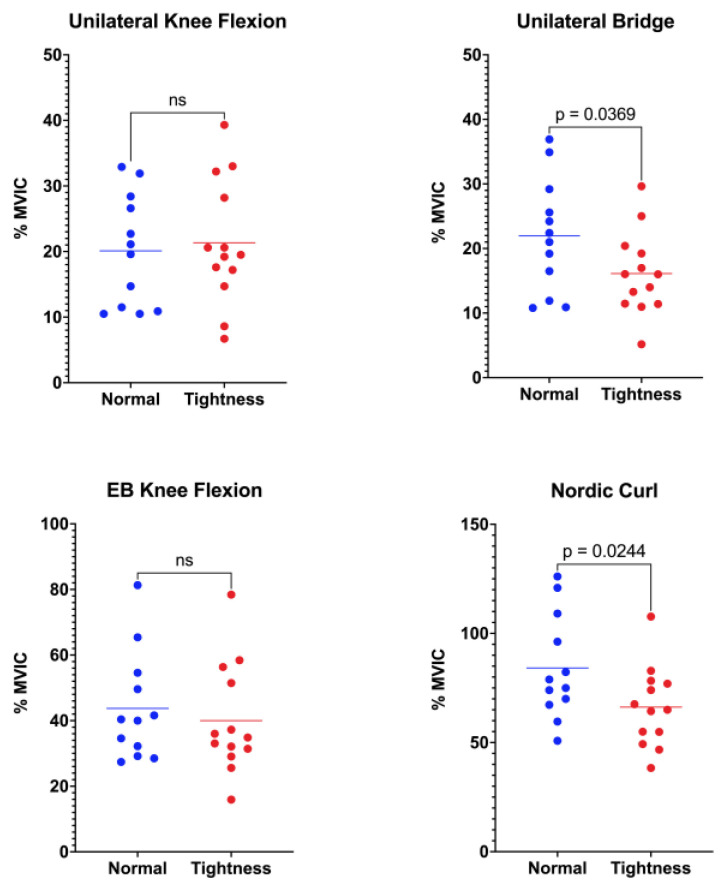
Comparison of semitendinosus muscle activation between the normal-flexibility and hamstring-tightness groups during the following exercises: unilateral knee flexion, unilateral bridge, elastic-band (EB) knee flexion, and Nordic curl. Statistical differences are indicated in the figure; “ns” denotes non-significant differences.

**Table 1 jfmk-10-00363-t001:** Characteristics of the sample.

	Normal		Tightness	
Variable	Mean	SD	Mean	SD
Age (Years)	21.08	2.1	21.46	2.18
Weight (Kg)	72.85	11.5	73.31	11.38
Height (m)	1.72	0.06	1.73	0.05
BMI (Kg/m^2^)	24.62	3.61	24.57	3.71
AKE Test (°)	11.75	4.39	32.31	5.36

BMI: body mass index; AKE: Active Knee Extension.

## Data Availability

The data presented in this study are available on request from the corresponding author (E.G.-M.). The data are not publicly available due to privacy and ethical restrictions.

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
