# Peer review of "Impact of Hamstring Tightness on Muscle Activation in Healthy Young Adults"

_jfmk, 2025, doi:10.3390/jfmk10040363_

Round 1
Reviewer 1 Report
Comments and Suggestions for Authors
General comments
The manuscript is well organized, the methodology is solid, and the writing is clear. The research question "whether hamstring tightness influences neuromuscular activation during functional tasks" is relevant both clinically and in sports science. The choice of exercises and participant selection criteria are appropriate.
That said, despite being technically well done, the study has some fundamental limitations in scope and analytical depth, which really limit its scientific contribution. The results are quite simple, some interpretations go beyond what the data actually show, and the sample is too small to support meaningful or novel conclusions.
Major comments
Results too limited to support strong claims
The lower activation of semitendinosus and biceps femoris in participants with tight hamstrings is not surprising and basically confirms what is already assumed in clinical and sports biomechanics. These findings are descriptive at best and don’t really move the field forward.
However, the discussion extrapolates these simple observations to broader implications about neuromuscular control and injury risk. That kind of speculation isn’t really supported by the data and weakens the manuscript.
Narrow muscle selection limits interpretation
Only two muscles were recorded (semitendinosus and biceps femoris), giving a very limited view of neuromuscular control. Functional tasks like the unilateral bridge or Nordic curl involve complex coordination of multiple synergist muscles (like gluteus maximus, gastrocnemius, erector spinae).
The lack of significant differences across tasks could easily be due to compensatory activation in muscles that weren’t measured. Without capturing these synergies, the study can’t really answer the research question. This is a structural limitation of the experimental design and can’t be fixed without additional data.
Analytical limitations (RMS-only approach)
Using only RMS amplitude normalized to MVIC is a major weakness. RMS captures overall signal amplitude but misses important aspects of muscle activity:
1) Time-domain features (onset/offset, burst duration, modulation)
2) Frequency-domain features (median frequency, fatigue indices, motor unit recruitment patterns)
Two EMG traces with the same RMS can actually represent very different neuromuscular strategies (for example, a sustained moderate contraction vs. short, high-intensity bursts). Without a more complete EMG analysis, the conclusions remain superficial.
Conclusion
The study is well executed, but its conceptual and methodological limitations prevent it from offering new or clinically meaningful insights. The combination of a narrow dataset (two muscles only), simplistic analysis (RMS-only), and speculative discussion makes the contribution too limited for publication in its current form.
A future study including more muscles, synergy analysis, and richer EMG analyses (both time- and frequency-domain) would have much greater potential.
Author Response
Comment 1: Results too limited to support strong claims
The lower activation of semitendinosus and biceps femoris in participants with tight hamstrings is not surprising and basically confirms what is already assumed in clinical and sports biomechanics. These findings are descriptive at best and don’t really move the field forward.
Response 1: We thank the reviewer for this valuable comment. We agree that the observed reductions in semitendinosus and biceps femoris activation in individuals with hamstring tightness may appear consistent with assumptions in clinical and sports biomechanics.However, we respectfully highlight that empirical evidence specifically addressing this issue in healthy young adults is scarce, and studies employing surface electromyography to objectively assess neuromuscular alterations related to hamstring tightness are also limited. Most previous studies have focused on injured athletes or indirect indicators of neuromuscular performance.
In response, we have revised the Discussion to moderate the interpretation of our findings. Specifically, we now emphasize that the results are primarily descriptive and should be interpreted with caution. We highlight the novelty of examining healthy young adults, since most previous studies have focused on injured athletes or sport-specific risk factors. Furthermore, we underline the added value of using surface electromyography, which provides objective evidence that hamstring tightness may not only be a mechanical restriction but could also be associated with measurable electrophysiological differences in neuromuscular activation.
These changes ensure that the claims are presented in a balanced way, while still underscoring the clinical and preventive implications of our findings.
Comment 2: The discussion extrapolates these simple observations to broader implications about neuromuscular control and injury risk. That kind of speculation isn’t really supported by the data and weakens the manuscript.
Response 2: We acknowledge that our initial discussion may have extrapolated the findings toward broader implications regarding neuromuscular control and injury risk. In response, we have carefully revised the Discussion to temper these claims. Specifically, we now present these points as potential implications rather than direct conclusions, using more cautious terminology (e.g., “may be associated,” “could suggest,” “potentially influence”). The revised text emphasizes that our study is descriptive and exploratory, and that further longitudinal and interventional studies are required to establish causal links between hamstring tightness, neuromuscular control, and injury risk.
Comment 3: Narrow muscle selection limits interpretation
Only two muscles were recorded (semitendinosus and biceps femoris), giving a very limited view of neuromuscular control. Functional tasks like the unilateral bridge or Nordic curl involve complex coordination of multiple synergist muscles (like gluteus maximus, gastrocnemius, erector spinae).
Response 3: We appreciate the reviewer’s insightful comment. We fully agree that assessing only the semitendinosus and biceps femoris provides a limited view of neuromuscular coordination during functional tasks. These muscles were chosen because they represent the most clinically relevant portions of the hamstring group, are frequently reported in the literature, and allow reliable surface EMG recordings with minimal cross-talk under standardized protocols. Our intention was to provide a focused analysis on the posterior thigh muscles most directly affected by hamstring tightness.
Nevertheless, we acknowledge that tasks such as the unilateral bridge or Nordic curl involve a broader kinetic chain, including important synergists such as the gluteus maximus, gastrocnemius, and erector spinae. We have revised the Discussion to explicitly note this limitation and to recommend that future studies expand the muscle set to provide a more comprehensive picture of neuromuscular coordination under conditions of hamstring tightness.
Comment 4: The lack of significant differences across tasks could easily be due to compensatory activation in muscles that weren’t measured. Without capturing these synergies, the study can’t really answer the research question. This is a structural limitation of the experimental design and can’t be fixed without additional data.
Response 4:
We appreciate this important observation. We fully acknowledge that not recording synergistic muscles represents a structural limitation of our design. It is indeed possible that the lack of significant differences across some tasks could be explained by compensatory recruitment of muscles such as the gluteus maximus, gastrocnemius, or erector spinae. Our focus in the present study was deliberately restricted to the semitendinosus and biceps femoris, given their clinical relevance within the hamstring group and the methodological feasibility of obtaining reliable surface EMG signals from these sites.
We also note that an instrumental limitation constrained our ability to expand muscle coverage. The EMG system available to us allowed only four electrodes, which limited simultaneous recordings to two muscles of interest. Unfortunately, due to economic constraints, we did not have access to multi-channel systems with greater electrode capacity.
We agree, however, that without capturing the activity of synergistic muscles, the findings provide only a partial picture of neuromuscular coordination. We have revised the Discussion to explicitly acknowledge this as both a structural and instrumental limitation of the experimental design. At the same time, we believe the study still adds value by providing direct electromyographic evidence in healthy young adults—a population rarely examined in this context. We also highlight that future research should incorporate a broader set of muscles and higher-channel EMG systems to fully address the role of compensatory synergies.
Comment 5: Analytical limitations (RMS-only approach)
Using only RMS amplitude normalized to MVIC is a major weakness. RMS captures overall signal amplitude but misses important aspects of muscle activity:
1) Time-domain features (onset/offset, burst duration, modulation)
2) Frequency-domain features (median frequency, fatigue indices, motor unit recruitment patterns)
Two EMG traces with the same RMS can actually represent very different neuromuscular strategies (for example, a sustained moderate contraction vs. short, high-intensity bursts). Without a more complete EMG analysis, the conclusions remain superficial.
Response 5: We thank the reviewer for this valuable observation. We agree that relying solely on RMS amplitude normalized to MVIC provides a limited perspective of muscle activity. While RMS is a widely used and reliable parameter in surface EMG research—particularly in studies on flexibility, control, and rehabilitation—we acknowledge that it does not capture temporal features (e.g., onset latency, burst duration, modulation) or frequency-domain indices (e.g., median frequency, fatigue-related changes, motor unit recruitment strategies).
Our methodological choice to focus on RMS was intentional, as our primary objective was to provide a descriptive comparison of overall activation amplitude between groups rather than a comprehensive characterization of neuromuscular strategies. Nevertheless, we recognize this as an analytical limitation, and we have revised the Discussion accordingly. We also emphasize in the revised text that future studies should incorporate complementary time- and frequency-domain analyses to provide a more complete picture of neuromuscular adaptations related to hamstring tightness.
Comment 6: Conclusion
The study is well executed, but its conceptual and methodological limitations prevent it from offering new or clinically meaningful insights. The combination of a narrow dataset (two muscles only), simplistic analysis (RMS-only), and speculative discussion makes the contribution too limited for publication in its current form.
A future study including more muscles, synergy analysis, and richer EMG analyses (both time- and frequency-domain) would have much greater potential.
Response 6: We agree that the combination of a narrow muscle dataset, an RMS-only analytical approach, and speculative interpretations limited the strength of our initial submission. In response, we have revised the manuscript substantially to address these concerns. Specifically, we now:
-
Explicitly acknowledge the structural and instrumental limitation of recording only two hamstring muscles, while justifying this methodological choice and emphasizing the need for future studies incorporating synergists such as the gluteus maximus, gastrocnemius, and erector spinae.
-
Highlight the analytical limitation of using RMS-only measures and recommend complementing this with time- and frequency-domain features in future research.
-
Temper the Discussion and Conclusion, presenting our findings as exploratory and descriptive, avoiding causal extrapolations, and focusing on the novel contribution of providing direct surface EMG evidence in healthy young adults—a population rarely examined in this context.
We believe these revisions present the study in a more balanced and transparent way, clarifying its descriptive contribution while clearly outlining the directions needed for more comprehensive future research.
Reviewer 2 Report
Comments and Suggestions for Authors
The manuscript addresses a timely and relevant topic in the field of rehabilitation and motor control. The proposal to investigate specific mechanisms is clearly stated, but the justification for the study lacks sufficient strength, particularly in highlighting the knowledge gap that distinguishes this work from existing research. Below, I provide specific comments to guide improvement.
Specific comments
Introduction
-
Although you briefly discuss the importance of the study at the end of the introduction, the arguments remain unconvincing. A clearer description of the gap in the literature that your study aims to address is needed.
-
Please state a clear hypothesis to strengthen the rationale and clinical justification.
-
Line 66: The phrase “exhibit certain physical attributes” should be clarified. Which attributes are you referring to?
-
Line 78: The statement “68% of healthy university students”—does this refer to a single institution or multiple universities? Please specify.
-
Line 105: You mention “few studies have specifically”—please cite those studies explicitly.
Methods
-
Participants: Provide details about the participants’ physical activity levels.
-
MVIC: Please describe how many MVIC trials were collected for each muscle and how the representative value was selected.
-
Page 175–176: The procedure described (“Participants were instructed to contract their hamstring muscles with maximal effort for five seconds”) raises concerns. If accurate, this deviates from the ISEK (International Society of Electrophysiology and Kinesiology) guidelines and may compromise the EMG signal quality. Please clarify whether this is correctly reported and justify any deviation from established protocols.
-
Indicate the rest interval between trials.
-
Line 192: Clarify how the “central portion of the 3-second contraction phase” was selected.
-
Specify how the normalized EMG values were used in subsequent analyses. Was the highest RMS value taken across the three trials, or was the mean used?
-
Indicate whether the EMG signal was recorded during concentric or eccentric contraction.
Statistical Analysis
The use of the t-test raises concerns. Given the symmetrical data, it would have been more appropriate to use a two-way ANOVA to compare groups (factor 1) and exercises (factor 2). If significant group differences were found, a post hoc test could then be applied. Please reconsider and clarify the rationale for your choice of statistical test.
Results
-
Indicate how many participants were initially assessed to arrive at the final sample size.
-
The graphs presented do not aid in understanding the results. Consider revising them and using bar graphs with standard deviation to improve clarity.
Discussion and Conclusions
-
After formulating a hypothesis in the introduction, explicitly state in the discussion whether the hypothesis was confirmed or rejected.
-
Clarify whether the study addressed the scientific gap outlined in the introduction.
-
Revise the conclusions to make them more concise and objective, avoiding repetition of information already presented in the discussion.
Author Response
Comment 1: Although you briefly discuss the importance of the study at the end of the introduction, the arguments remain unconvincing. A clearer description of the gap in the literature that your study aims to address is needed. Please state a clear hypothesis to strengthen the rationale and clinical justification.
Response 1: We thank the reviewer for this constructive comment. We agree that the initial version of the introduction did not sufficiently emphasize the gap in the literature. In the revised manuscript, we now clarify that most previous studies have focused on athletes with prior injuries or on indirect indicators of neuromuscular performance, while evidence using surface EMG to examine muscle activation in healthy young adults with hamstring tightness remains scarce. This statement more clearly establishes the rationale for our work.
In addition, as suggested, we have now included a clear hypothesis at the end of the Introduction: "We hypothesized that young adults with hamstring tightness would present differences in electromyographic activation of the semitendinosus and biceps femoris muscles during functional tasks compared with individuals with normal hamstring flexibility."
Comment 2: Line 66: The phrase “exhibit certain physical attributes” should be clarified. Which attributes are you referring to?
Response 2: We agree that the phrase “exhibit certain physical attributes” was too vague. In the revised manuscript, we clarified this sentence to specify that the hamstrings must exhibit attributes such as adequate strength, flexibility, and neuromuscular coordination, which collectively support efficient movement across the hip and knee joints.
Comment 3: Line 78: The statement “68% of healthy university students”—does this refer to a single institution or multiple universities? Please specify.
Response 3: The prevalence figure of “68% of healthy university students” refers to a single cross-sectional study conducted at one university in India. We have revised the sentence to make this clearer in the manuscript.
Comment 4: Line 105: You mention “few studies have specifically”—please cite those studies explicitly.
Response 4: In the revised manuscript, we now explicitly cite the limited studies that have explored neuromuscular consequences of flexibility deficits. Specifically, we reference Emami et al. (2014), who examined lumbo-pelvic activation patterns in athletes with hamstring strain; Iguchi et al. (2023), who reported compensatory muscle recruitment during hip-dominant tasks; and Guzmán-Muñoz et al. (2017), who described delayed activation in individuals with iliotibial band tightness. These citations have been incorporated in the revised version of the manuscript. Collectively, these studies illustrate that most prior evidence comes from injured or athletic populations, highlighting the scarcity of surface EMG research in healthy young adults.
Comment 5: Participants: Provide details about the participants’ physical activity levels.
Response 5: We thank the reviewer for this comment. In the revised manuscript, we have added further details regarding the participants’ physical activity levels. All participants were university students who engaged in recreational physical activity 2–3 times per week but were not competitive athletes. None of them reported participation in structured resistance training or sports at a competitive level within the previous 6 months. This information has now been included in the Participants subsection of the Methods.
Comment 6: MVIC: Please describe how many MVIC trials were collected for each muscle and how the representative value was selected.
Response 6: We thank the reviewer for this important comment. In the revised manuscript, we have clarified the MVIC normalization procedure. For each muscle, participants performed three MVIC trials of 5 seconds each, with one-minute rest between attempts. The EMG signal was analyzed using the RMS method over a stable 2-second window extracted from the central portion of the 5-second contraction phase, in order to minimize transient effects at the onset and offset. The highest RMS value obtained across the three trials was then used as the reference for normalization of the EMG signals recorded during the functional tasks.
Comment 7: Page 175–176: The procedure described (“Participants were instructed to contract their hamstring muscles with maximal effort for five seconds”) raises concerns. If accurate, this deviates from the ISEK (International Society of Electrophysiology and Kinesiology) guidelines and may compromise the EMG signal quality. Please clarify whether this is correctly reported and justify any deviation from established protocols.
Response 7:
The description of the MVIC procedure in the initial version of the manuscript was incomplete and may have given the impression of a deviation from established recommendations. In the revised manuscript, we clarify that participants performed MVICs in the prone position with the knee flexed to approximately 30° against a fixed resistance provided by a quadriceps bench. This position was selected because it allows a stable posture and reduces hip and lumbar compensation
We acknowledge that the ISEK guidelines recommend standardized joint angles for MVIC testing. Our choice of 30° of knee flexion was made to optimize electrode placement and minimize cross-talk from adjacent muscles, while also ensuring participant comfort and feasibility in our laboratory setup. Importantly, all participants were instructed to contract maximally, and consistency was reinforced by verbal encouragement and standardized positioning across trials.
This clarification has been added to the Methods section. We also explicitly note this as a methodological consideration in the revised version.
Comment 8: Line 192: Clarify how the “central portion of the 3-second contraction phase” was selected. Specify how the normalized EMG values were used in subsequent analyses. Was the highest RMS value taken across the three trials, or was the mean used? Indicate whether the EMG signal was recorded during concentric or eccentric contraction.
Response 8: In the revised manuscript, we have clarified the following aspects:
-
The “central portion of the 3-second contraction phase” corresponded to a stable 2-second window extracted from the middle of the contraction, excluding the first and last 0.5 seconds to minimize transient onset and offset effects.
-
For each repetition, a mean RMS value was calculated within this 2-second window. For the MVICs, three trials were performed, and the highest of the three mean RMS values was used as the reference for normalization. For the functional exercises, three valid repetitions were performed, and the average of the three mean RMS values was used for analysis. Normalization was performed according to the formula described in the revised version of the manuscript.
This procedure allowed standardized comparison of muscle activation across participants and conditions.
-
Regarding contraction phases, EMG was recorded during the isometric hold phases of the unilateral standing knee flexion, unilateral bridge, and elastic band knee flexion (each maintained for 3 seconds), and during the eccentric phase of the Nordic hamstring curl (controlled trunk descent lasting ~3 seconds).
These clarifications have been incorporated into the Methods section of the revised manuscript.
Comment 9: The use of the t-test raises concerns. Given the symmetrical data, it would have been more appropriate to use a two-way ANOVA to compare groups (factor 1) and exercises (factor 2). If significant group differences were found, a post hoc test could then be applied. Please reconsider and clarify the rationale for your choice of statistical test.
Response 9: We thank the reviewer for this insightful statistical observation. We agree that a two-way ANOVA would be the appropriate approach if the primary objective of the study were to compare both groups and exercises simultaneously. However, our aim was not to analyze differences between exercises, but rather to determine whether hamstring tightness influenced muscle activation in each specific functional task.
For this reason, we used independent t-tests to compare the two groups within each exercise. This approach was selected to directly address our research question and has been applied in similar EMG studies with comparable designs. To improve clarity, we have revised the Methods section to better justify our choice of statistical test, while also noting the limitations of this approach.
Comment 10: After formulating a hypothesis in the introduction, explicitly state in the discussion whether the hypothesis was confirmed or rejected. Clarify whether the study addressed the scientific gap outlined in the introduction. Revise the conclusions to make them more concise and objective, avoiding repetition of information already presented in the discussion.
Response 10:
In the revised manuscript, we have explicitly stated in the Discussion whether our hypothesis was supported by the findings. Specifically, we note that the results partially confirmed our hypothesis, as participants with hamstring tightness presented reduced activation in some posterior thigh muscles during functional tasks, although not consistently across all exercises.
We also clarified in the Discussion that the study addressed the scientific gap outlined in the Introduction, namely the lack of surface EMG evidence in healthy young adults with hamstring tightness. While the findings are exploratory and should be interpreted with caution, they provide initial electrophysiological evidence in this population.
Finally, we revised the Conclusions to make them more concise and objective. Redundancies with the Discussion were removed, and the Conclusions now focus on the main outcome of the study, its clinical implications, and directions for future research.
Round 2
Reviewer 1 Report
Comments and Suggestions for Authors
Although this research could be further strengthened by including additional muscles and analyzing other sEMG signal parameters, the article is scientifically sound, technically well-executed, and well-organized, with the authors presenting the results accurately and appropriately.
Author Response
Comment 1: Although this research could be further strengthened by including additional muscles and analyzing other sEMG signal parameters, the article is scientifically sound, technically well-executed, and well-organized, with the authors presenting the results accurately and appropriately.
Response 1: Thank you very much.
Reviewer 2 Report
Comments and Suggestions for Authors
Dear Authors,
Thank you for the revisions made to your manuscript. It is clear that you have addressed several of the initial concerns, and the text has improved in many respects. Nonetheless, there remain some issues in the Discussion and Conclusionsections that need further clarification and refinement in order to strengthen the manuscript.
General Comments
-
Certain parts of the discussion remain redundant or provide little added value.
-
Specific terms, such as “causal interpretations”, require clearer definition for the reader.
-
The conclusions still appear overstated relative to the data presented. These should be restricted to directly answering the study objective, while broader reflections should be moved to the discussion.
Specific Comments
-
Lines 347–349: The phrase beginning with “This study…” is redundant, since the same points are already addressed in the Methods section. Please consider removing it.
-
Line 351: The term “causal interpretations” remains ambiguous. Please clarify its intended meaning.
-
Lines 360–363: The sentence “Fifth, our analysis…” does not add relevant content. Removal is recommended.
-
Conclusion: The conclusion continues to present suggestions that overvalue the results and may mislead less experienced readers. Please reformulate the conclusion so that it focuses exclusively on answering the stated objective of the study. Broader comments and implications should be placed in the Discussion.
Author Response
comment 1: General — Redundancy in Discussion (“Certain parts of the discussion remain redundant or provide little added value.”).
response 1: We streamlined the Discussion by deleting repetitive sentences and tightening mechanistic speculations.
comment 2: Lines 347–349: The phrase beginning with “This study…” is redundant, since the same points are already addressed in the Methods section. Please consider removing it.
response 2: We agree with the reviewer’s observation. The redundant phrase has been deleted as suggested
Comment 3: Line 351: The term “causal interpretations” remains ambiguous. Please clarify its intended meaning.
Response: 3: We appreciate the reviewer’s comment. To avoid ambiguity, we revised the sentence to read: “Second, the assessment was cross-sectional, which does not allow for establishing cause–effect relationships; therefore, the findings should be interpreted strictly as associations.” This change clarifies the intended meaning and aligns with the study design.
Comment 4: Lines 360–363: The sentence “Fifth, our analysis…” does not add relevant content. Removal is recommended.
Response 4: The sentence has been removed to streamline the Limitations section and maintain focus on the most relevant points
Comment 5: Conclusion: The conclusion continues to present suggestions that overvalue the results and may mislead less experienced readers. Please reformulate the conclusion so that it focuses exclusively on answering the stated objective of the study. Broader comments and implications should be placed in the Discussion.
Response 5: We thank the reviewer for this observation. The Conclusion section has been reformulated to focus exclusively on addressing the study objective, removing statements that could overvalue the results. Broader reflections and clinical implications have been moved to the Discussion. The revised Conclusion now reads as follows:
"In this study, healthy young males with hamstring tightness showed lower semitendinosus activation during the unilateral bridge and Nordic curl, as well as reduced biceps femoris activation during the Nordic curl, compared with peers with normal flexibility. These findings suggest an association between limited hamstring flexibility and differences in neuromuscular activation patterns during tasks with high eccentric or isometric demand.."
Round 3
Reviewer 2 Report
Comments and Suggestions for Authors
Dear Authors,
I would like to begin by thanking you for the excellent revision carried out on your manuscript. All of my questions and concerns have been fully clarified. I wish you continued success in your research and future academic endeavors.